# GeoBA: Stealthy Geometric Poisoning on 3D Point Cloud

## Abstract

Point cloud backdoor attacks exploit carefully crafted trigger patterns to manipulate deep neural networks (DNNs), causing misclassification when specific input patterns are encountered. Existing approaches primarily rely on (1) explicit trigger injection (e.g., adding a specific shape) or (2) basic geometric transformations (e.g., rotation, scaling) to generate poisoned samples. However, such trigger patterns are often easily detected by the human eye or statistical analysis, undermining the stealth and effectiveness of the attack. To this end, we propose GeoBA, a stealthy geometric poisoning backdoor attack that embeds imperceptible yet robust triggers into point clouds with minimal geometric perturbation. Specifically, we first transform point clouds into a spherical domain, where subtle phase perturbations are applied to introduce the backdoor pattern while preserving the global geometric structure. This perturbation effectively induces the model to learn the trigger while avoiding noticeable shape deviations. A controlled inverse transformation then maps the poisoned samples back to the original space, ensuring their imperceptibility and robustness to existing defenses. Experiments show that GeoBA consistently triggers backdoors across mainstream 3D architectures (e.g., Mamba3D, PointMLP), with excellent stealth, transferability, and robustness—highlighting overlooked security risks in geometric transformations. Excitingly, it only takes 4 lines of core code to achieve this. *The code will be released promptly.*

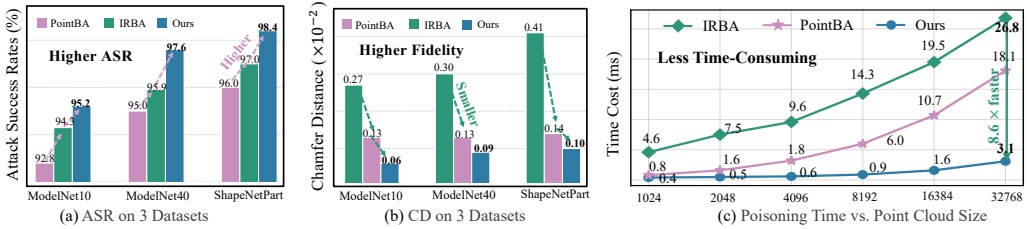

Figure 1: Comparison of ASR, CD, and runtime of PointBA (ball trigger), IRBA (scaling trigger), and our GeoBA trigger across three datasets. Our method outperforms in multiple dimensions, with results highlighted in *bold*.

## 1 Introduction

The widespread use of 3D point cloud technology in autonomous driving Li et al. (2020); Chen et al. (2017a); Hu et al. (2022) and robotics Pfrunder et al. (2017); Oh et al. (2024) has made it a prime target for backdoor attacks. Existing methods Chen et al. (2017b); Nguyen & Tran (2021); Li et al. (2021b) often adapt 2D attack strategies without leveraging the geometric properties of point clouds, limiting stealth and effectiveness. As shown in Figure 1, our geometry-aware approach balances attack success rate (ASR), shape fidelity, and computational efficiency, addressing key gaps in practical deployment. Current 3D backdoor attacks fall into two main categories Li et al. (2021a); Xiang et al. (2021); Bian et al. (2024): the first, exemplified by PointBA Li et al. (2021a), injects explicit triggers by adding geometric structures, disrupting the native topology like grafting

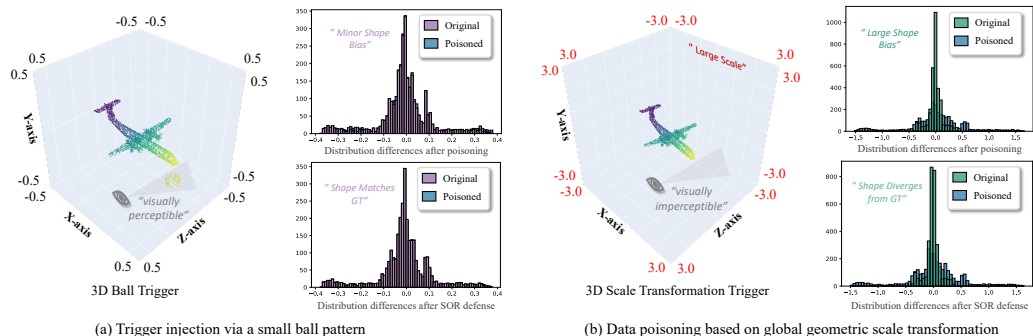

(a) Trigger injection via a small ball pattern  (b) Data poisoning based on global geometric scale transformation

Figure 2: Visual analysis of backdoor attack paradigms: (a) Ball trigger shows statistical stealth but is significantly weakened after preprocessing (e.g., SOR); (b) Scaling-based trigger exhibits perceptual stealth and robustness to SOR, yet introduces noticeable distributional shifts.

an artificial limb. Figure 2(a) demonstrates that geometric purification methods such as SOR Zhou et al. (2019) effectively smooth these triggers, neutralizing the attack and revealing its vulnerability.

The second category employs global geometric transformations (e.g., rotation, scaling, or nonlinear deformation) Li et al. (2021a); Gao et al. (2023); Wang et al. (2024) to perturb point clouds without explicit trigger insertion. By exploiting intrinsic geometric properties, these methods maintain structural integrity and offer flexibility across diverse datasets and architectures. Such transformations preserve attack efficacy without relying on dataset-specific patterns, thereby enhancing transferability and robustness. However, their coarse-grained and uniform nature often leads to noticeable shape distortions, as illustrated in Figure 2(b), compromising geometric fidelity and increasing detectability. This exposes a key challenge in 3D backdoor design: how to manipulate the latent feature space effectively while avoiding perceptible geometric alterations.

Inspired by ripple dynamics Sinha et al. (2016); Wen et al. (2024); Liu et al. (2022), we observe that spectral functions modulate amplitude to preserve global structure while phase shifts induce local radial deformations. Building on this insight, spectral perturbations serve as triggers in point clouds, enabling controllable local shifts without perceptible geometric changes. However, the irregularity of 3D point clouds poses two key challenges for spectral transformations: (1) local deformation disrupts isometry, causing geometric distortion; (2) phase perturbations induce scale drift, compromising spatial consistency. This raises our central question: *Can we design a backdoor mechanism that respects point cloud geometry while enabling spectral controllability?*

To address the above challenges, we propose GeoBA, a stealthy geometric poisoning backdoor attack that embeds imperceptible yet robust triggers into point clouds with minimal geometric distortion. GeoBA forgoes explicit trigger insertion and global geometric transformations, instead employing phase-aware poisoning within the intrinsically constrained spherical domain to realize backdoor attacks. Specifically, GeoBA injects phase triggers in the tangential space via spherical mapping, preserving radial distances to maintain geometry while embedding backdoors. Then, an inverse transformation maps the perturbed point cloud back to the original space, ensuring imperceptibility to humans and robustness against common purification defenses (e.g., SOR Zhou et al. (2019)). Extensive experiments across multiple 3D architectures demonstrate that GeoBA achieves state-of-the-art stealthiness, transferability, and resilience, revealing critical security risks in geometric transformations and offering new perspectives on backdoor attack design in 3D deep learning. In summary, the main contributions of this paper are as follows:

1. We propose GeoBA, a novel backdoor attack that embeds imperceptible triggers into point clouds via spatially minimal geometric distortion.

2. Leveraging phase perturbations in the spherical domain, GeoBA stealthily embeds backdoors while preserving overall geometric integrity.

3. Extensive experiments on various 3D architectures show GeoBA's state-of-the-art stealth and robustness against preprocessing defenses.

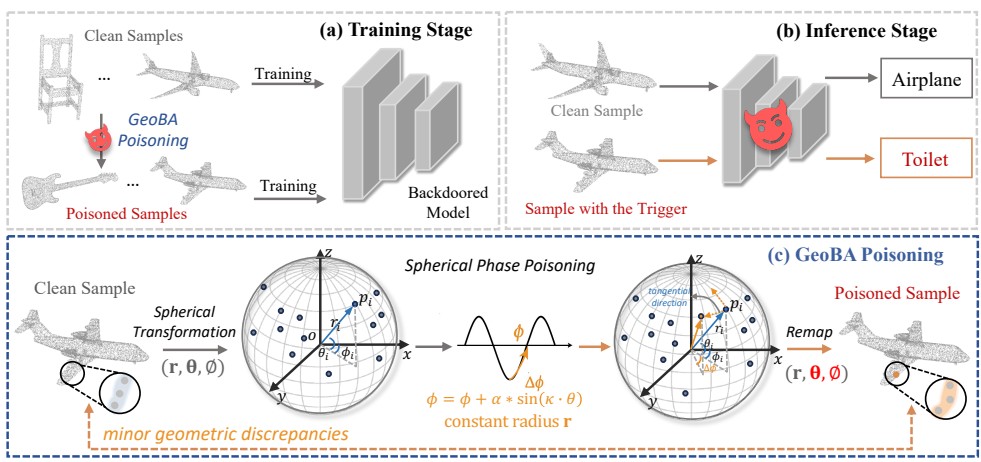

Figure 3: GeoBA's clean-label backdoor attack pipeline. (a) Training stage: Poisoned samples (marked with spherical phase perturbations $\varphi = \varphi + \alpha \cdot \sin(\kappa\theta)$) are injected into the training set. (b) Inference stage: The backdoored model misclassifies triggered samples (e.g., airplane→toilet) while maintaining accuracy on clean inputs. Bottom: The spherical coordinate transformation achieves imperceptible perturbations through phase-angle modulation along tangential displacements.

## 2 RELATED WORK

### 2.1 3D POINT CLOUD CLASSIFICATION

Deep learning has revolutionized 3D point cloud processing, with mainstream methods falling into four categories: point-wise MLP Ma et al. (2022), convolution-based Li et al. (2018), graph-based Wang et al. (2019), and transformer-based Guo et al. (2021) approaches. PointNet Qi et al. (2017a) pioneered this field using shared MLPs and max-pooling for permutation invariance, while PointNet++ Qi et al. (2017b) extended it with hierarchical local feature aggregation. Alternative approaches include PointCNN's order-aware X-convolutions and DGCNN's dynamic graph networks for local structure modeling. Recent architectures, including Transformer-based PTv3 Wu et al. (2024) and state-space model series (PCM Zhang et al. (2025b), CamPoint Zhang et al. (2025a), Mamba3D Han et al. (2024)), have demonstrated state-of-the-art performance in point cloud recognition and segmentation tasks.

Despite their success in point cloud tasks, these models remain underexplored in terms of backdoor vulnerability. This security gap calls for systematic evaluation on representative architectures (e.g., PointNet, PointNet++, DGCNN, and PCT) to assess backdoor and guide robust model design.

### 2.2 BACKDOOR ATTACKS IN 3D POINT CLOUD

Backdoor attacks on 3D point cloud classifiers Gao et al. (2023); Li et al. (2021a; 2022); Wei et al. (2024); Feng et al. (2025); Bian et al. (2024) manipulate model predictions by embedding malicious triggers into training data. Early approaches, such as PointBA-Ball Li et al. (2021a); Xiang et al. (2021), rely on explicit 3D triggers (e.g., additional clustered points) to execute the attack, while later works explore geometric transformation-based triggers, including rigid rotations PointAPA Wang et al. (2024) and PointBA-O Li et al. (2021a). More recent efforts employ nonlinear deformations IRBA Gao et al. (2023) and noise-based modifications Fan et al. (2024) to improve stealth, yet these methods remain vulnerable to statistical outlier removal (SOR) Zhou et al. (2019) and standard data augmentations. Additionally, iBA Bian et al. (2024) and SPBA Feng et al. (2025) introduce dedicated trigger networks for implicit injection, enhancing stealth while increasing computation.

By contrast, GeoBA exploits the intrinsic constraints of spherical space to construct a flexible, non-parametric backdoor strategy that embeds stealthy triggers while preserving global geometry, achieving both macroscopic imperceptibility and microscopic controllability with strong robustness against pre-training data preprocessing.

## 3 THREAT MODEL

We consider a practical poison-label backdoor attack where an adversary contaminates a third-party point cloud dataset without access to downstream models or training details [6,8,19]. The attacker is limited by three constraints: (1) poisoning less than 5% of training samples, (2) no changes to model architectures or training pipelines, (3) Users typically cannot detect whether data has been poisoned or identify the poisoning type, and (4) poisoned samples must withstand common preprocessing such as augmentation and denoising Borgnia et al. (2021); Wang et al. (2024); Gao et al. (2023). The attack succeeds if models trained on the poisoned data misclassify triggered inputs as a target label while preserving accuracy on clean samples. Importantly, geometric perturbations must maintain visual fidelity to evade human and automated detection.

The threat model mainly includes two categories: (1) parameter-free data poisoning (e.g., embedding triggers via distortion or rotation); and (2) proxy-based prediction injection (e.g., inserting geometric triggers). Later methods (PointAPA, SPBA, IBA, etc.) are extensions of these two paradigms.

## 4 METHODOLOGY

### 4.1 PRELIMINARIES

In the context of 3D deep learning, a point cloud classifier is trained to map a set of 3D point coordinates to a predefined label space. Formally, let the training dataset be defined as:

$$\mathcal{D} = \{(\boldsymbol{P}_i, \boldsymbol{Y}_i)\}_{i=1}^N, \quad \boldsymbol{P}_i \in \mathbb{R}^{K \times 3}, \quad \boldsymbol{Y}_i \in \{1, 2, \ldots, C\},$$

where $\boldsymbol{P}_i$ represents a point cloud instance containing $K$ individual points, and $\boldsymbol{Y}_i$ is its corresponding ground-truth label from a total of $C$ possible classes. A deep learning-based classifier $f_{\boldsymbol{\theta}} : \mathbb{R}^{K \times 3} \to \{1, 2, \ldots, C\}$ is trained with model parameters $\boldsymbol{\theta}$, minimizing a loss function $\mathcal{L}(\cdot, \cdot)$ over the dataset $\mathcal{D}$.

In a backdoor attack, an adversary aims to implant a stealthy trigger into the dataset so that the trained model $f_{\boldsymbol{\theta}}$ behaves normally on clean inputs but misclassifies samples containing the trigger into a predefined target label $\boldsymbol{Y}_t$. To achieve this, the dataset is partitioned into two subsets:

$$\mathcal{D}_c = \{(\boldsymbol{P}_i, \boldsymbol{Y}_i)\}_{i=1}^{N-M}, \quad \mathcal{D}_b = \{(\hat{\boldsymbol{P}}_i, \boldsymbol{Y}_t)\}_{i=1}^M,$$

where $\mathcal{D}_c$ is the clean subset and $\mathcal{D}_b$ is the poisoned subset, consisting of a fraction $\rho = M/N$ of the dataset. The poisoned samples $\hat{\boldsymbol{P}}_i$ are generated by applying a transformation function $\mathcal{T}(\cdot)$, which modifies clean samples to embed a trigger while preserving the global structure:

$$\hat{\boldsymbol{P}}_i = \mathcal{T}(\boldsymbol{P}_i), \quad \forall \boldsymbol{P}_i \in \mathcal{D}_b.$$

Since real-world training pipelines commonly incorporate data augmentation and preprocessing steps such as Statistical Outlier Removal (SOR) and geometric transformations $\mathcal{P}(\cdot)$, an effective backdoor attack must ensure that the trigger remains functional even after such modifications:

$$f_{\boldsymbol{\theta}}(\mathcal{P}(\hat{\boldsymbol{P}}_i)) = \boldsymbol{Y}_t, \quad \forall \hat{\boldsymbol{P}}_i \in \mathcal{D}_b,$$

where $\mathcal{P}(\cdot)$ represents these preprocessing transformations. The objective of a successful backdoor attack is thus twofold: maximizing the **Attack Success Rate (ASR)**, defined as the proportion of poisoned samples classified as $\boldsymbol{Y}_t$, while minimizing the degradation in **Benign Accuracy (ACC)**, i.e., accuracy on clean samples:

$$\max_{\boldsymbol{\theta}} \sum_{\hat{\boldsymbol{P}}_i \in \mathcal{D}_b} \mathbb{I}\big(f_{\boldsymbol{\theta}}(\mathcal{P}(\hat{\boldsymbol{P}}_i)) = \boldsymbol{Y}_t\big), \quad \text{s.t.} \quad \text{ACC} \approx \text{ACC}_{\text{clean}},$$

where $\mathbb{I}(\cdot)$ is the indicator function.

This formulation highlights the challenge of designing an imperceptible yet robust backdoor trigger, particularly in 3D point clouds where spatial transformations and structural integrity play a crucial role in model generalization.

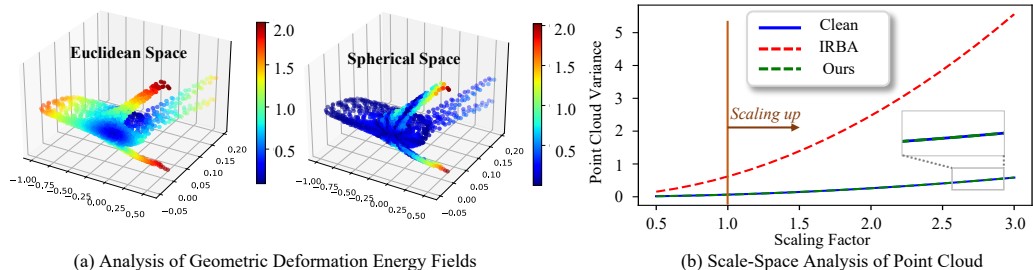

(a) Analysis of Geometric Deformation Energy Fields      (b) Scale-Space Analysis of Point Cloud

Figure 4: Visual comparison of euclidean-space (IRBA) poisoning vs. spherical phase-angle (GeoBA) poisoning.

## 4.2 SPHERICAL-GEOMETRIC POISONING

In this section, we present a detailed analysis of the proposed GeoBA backdoor attack. As shown in Figure 3, our goal is to generate robust poisoned samples that remain effective under various preprocessing techniques and different model architectures. These samples are embedded into the training phase and persist during inference, ensuring the backdoor is successfully activated.

**Euclidean Geometry vs. Spherical Geometry.** While most existing point cloud attacks operate in Euclidean space, they face fundamental challenges in 3D backdoor design: achieving subtle poisoning without causing noticeable shape distortion. As illustrated in Figure 4(a), conventional approaches like IRBA that employ linear transformation matrices $\mathbf{T} \in \mathbb{R}^{3 \times 3}$ suffer from:

- Global deformation propagation (left heatmap in Figure 4(a) left)
- Destruction of intrinsic scale properties (red dashed curve in Figure 4(b))

Motivated by the natural properties of spherical geometry for localized perturbation and scale preservation, we propose a novel poisoning method based on spherical coordinate transformation.

Given a point cloud $P = \{\boldsymbol{P}_i\}_{i=1}^N$, we first map each point into the spherical coordinate system:

$$(r_i, \theta_i, \phi_i) = \mathcal{S}(\boldsymbol{P}_i), \quad \mathcal{S} : \mathbb{R}^3 \to \mathbb{R}^+, \tag{1}$$

where $\mathcal{S}$ denotes the Cartesian-to-spherical transformation. Unlike global linear transformations in Euclidean space, we introduce phase-based perturbations within the tangent space:

$$\phi_i' = \phi_i + \delta_\phi \cdot \underbrace{\sin(k\theta_i)}_{\text{Polar Modulation}}, \tag{2}$$

This design offers two key advantages:

- **Locality Guarantee**: The modulation term $\sin(k\theta_i)$ concentrates the perturbation energy within specific polar bands (see right panel of Figure 4(a), cool-color regions with minimal hot spots), satisfying:

$$\|\nabla_\theta \delta_\phi \sin(k\theta)\| \le k\delta_\phi. \tag{3}$$

- **Geometric Preservation**: Both the radial distance $r_i$ and polar angle $\theta_i$ remain unchanged, thereby preserving the differential geometric structure of the point cloud.

Besides, under a uniform scaling transformation $\mathcal{X}_s(\boldsymbol{P}) = s\boldsymbol{P}$, Euclidean poisoning exhibit a scale-sensitive variance increase:

$$\text{Var}(\mathcal{T}_{\text{euc}}(\mathcal{X}_s(\boldsymbol{P}))) = s^4 \cdot \text{Var}(\mathcal{T}_{\text{euc}}(\boldsymbol{P})). \tag{4}$$

In contrast, our spherical poisoning maintains scale covariance:

$$\text{Var}(\mathcal{T}_{\text{sph}}(\mathcal{X}_s(\boldsymbol{P}))) = s^2 \cdot \text{Var}(\boldsymbol{P}), \tag{5}$$

this stems from the spherical mapping property:

$$\mathcal{S}(\mathcal{X}_s(\boldsymbol{P})) = (sr, \theta, \phi), \tag{6}$$

---

**Algorithm 1:** Geometric Poisoning on Point Clouds

---

**Input** : $\boldsymbol{P} \in \mathbb{R}^{N \times 3}$, phase shift $\delta_\phi$, frequency $k$
**Output:** Perturbed point cloud $\boldsymbol{P}_{\text{perturbed}} \in \mathbb{R}^{N \times 3}$

$(r, \theta, \phi) \leftarrow \text{CartesianToSpherical}(\boldsymbol{P})$                    // Convert coordinates
$\phi \leftarrow [\phi + \delta_\phi \cdot \sin(k \cdot \theta)] \bmod 2\pi$             // Perturb azimuthal angle
$\boldsymbol{P}_{\text{poisoned}} \leftarrow \text{SphericalToCartesian}(r, \theta, \phi)$        // Reconstruct point cloud
**return** $\boldsymbol{P}_{\text{poisoned}}$                                   // Output result

---

which naturally decouples the phase attack $\delta_\phi \sin(k\theta)$ from scale transformations. As shown by the nearly constant green curve in Figure 4(b), the attack maintains scale invariance since the scale factor $s$ exclusively modifies the radial component $r$, while our phase modulation depends solely on angular coordinates $\theta$ and $\phi$.

Therefore, our core methodology leverages spherical geometric properties to generate poisoned point cloud data, injecting latent backdoors during 3D network training while maintaining stealth. The detailed implementation is presented in Algorithm 1.

**Spherical Transformation.** Given a point cloud $\boldsymbol{P} \in \mathbb{R}^{N \times 3}$ with Cartesian coordinates $\boldsymbol{P}_i = (x_i, y_i, z_i)$, we transform each point to spherical coordinates $(r_i, \theta_i, \phi_i)$ via the nonlinear mapping:

$$\begin{cases} r_i = \sqrt{x_i^2 + y_i^2 + z_i^2} \\ \theta_i = \arccos(z_i/r_i) \\ \phi_i = \arctan 2(y_i, x_i) \end{cases} \tag{7}$$

where $r_i$ denotes the radial distance from origin, $\theta_i \in [0, \pi]$ represents the polar angle measuring inclination from the positive z-axis, and $\phi_i \in (-\pi, \pi]$ is the azimuthal angle in the xy-plane measured from the positive x-axis.

This bijective transformation establishes a complete mapping between Cartesian space and spherical manifold while preserving geometric information. The spherical representation naturally encodes rotational invariance in the angular domain, which proves particularly advantageous for 3D point cloud processing.

**Spherical Phase Poisoning.** This method introduces controlled perturbations to the azimuthal angle $\phi$ through harmonic phase modulation. Specifically, we apply a nonlinear transformation to the original spherical coordinates $(r_i, \theta_i, \phi_i)$, updating $\phi_i$ as $\phi_i' = \phi_i + \alpha \sin(k\theta_i)$ followed by modulo $2\pi$ to maintain periodicity. Here, $\alpha$ controls the poisoning intensity and $k$ governs the harmonic frequency. The sinusoidal term generates $\theta$-dependent phase shifts, ensuring structured yet subtle angular distortions. This strategy selectively perturbs the cyclic orientation information in the xy-plane while preserving radial distances. Key advantages include: (1) maintaining geometric plausibility via cyclic boundaries, (2) producing non-uniform perturbations that enhance stealthiness, and (3) retaining the original point density distribution. Hyperparameters $\alpha$ and $k$ are ablated in the experiments section.

**Spherical-Cartesian Remap.** The poisoned spherical coordinates $(r_i, \theta_i, \phi_i')$ are remapped to Cartesian space using the standard transformation: $x_i' = r_i \sin\theta_i \cos\phi_i'$, $y_i' = r_i \sin\theta_i \sin\phi_i'$, and $z_i' = r_i \cos\theta_i$. This reconstruction preserves the poisoned angular distribution while maintaining geometric validity. The backdoor threat remains effective for three key reasons: *First*, 3D DNNs fundamentally depend on position angular relationships between points for feature extraction. *Second*, the $\phi$-poisoning introduces consistent angular shifts that propagate through the network's feature space. *Third*, the preservation of radial distances $(r_i)$ maintains surface topology while only perturbing angular coordinates.

Crucially, our spherical-phase poisoning fundamentally differs from conventional Euclidean-space poisoning. Rather than direct coordinate manipulation, it targets implicit angular relationships in the non-Euclidean spherical manifold. The attack preserves exact pairwise Euclidean distances while creating geometrically plausible samples that systematically mislead deep classifiers. The remapped points $\boldsymbol{P}'$ maintain visual authenticity but contain learned angular perturbations that induce targeted misclassification, demonstrating that 3D DNNs are particularly sensitive to these spherical-space perturbations despite their Euclidean invariance.

| Victim Model | PointBA-I* | | PointBA-O | | IRBA | | **GeoBA (Ours)** | |
|---|---|---|---|---|---|---|---|---|
| | ACC↑ | ASR↑ | ACC↑ | ASR↑ | ACC↑ | ASR↑ | ACC↑ | ASR↑ |
| PointNet | 89.6 | 99.8 | 88.7 | 78.2 | 87.1 | 93.2 | 88.2 | 93.8 |
| PointNet++ | 91.3 | 98.8 | 91.0 | 91.3 | 90.5 | 95.7 | 91.0 | 97.3 |
| DGCNN | 90.6 | 100 | 91.0 | 82.0 | 91.3 | 94.0 | 91.2 | 96.7 |
| PointCNN | 91.2 | 100 | 91.2 | 82.9 | 83.9 | 94.6 | 90.2 | 93.6 |
| PCT | 90.6 | 100 | 90.5 | 81.7 | 89.5 | 74.2 | 90.4 | 95.0 |
| Avg | **90.7** | **99.7** | 90.5 | 83.2 | 88.5 | 90.3 | 90.2 | 95.3 |

Table 1: ACC (%) and ASR (%) of backdoored models with the PointBA-I (ball), PointBA-O (rotation), IRBA (scaling), and our GeoBA triggers on ModelNet40. Results of our proposed GeoBA are highlighted in *bold*.

| Victim Model | PointBA-I* | | PointBA-O | | IRBA | | **GeoBA (Ours)** | |
|---|---|---|---|---|---|---|---|---|
| | ACC↑ | ASR↑ | ACC↑ | ASR↑ | ACC↑ | ASR↑ | ACC↑ | ASR↑ |
| PointNet | 98.5 | 100 | 98.3 | 92.2 | 98.2 | 91.3 | 98.4 | 98.3 |
| PointNet++ | 98.9 | 100 | 98.6 | 92.6 | 98.9 | 99.0 | 98.5 | 99.0 |
| DGCNN | 98.8 | 100 | 98.9 | 88.9 | 98.7 | 83.0 | 98.8 | 99.3 |
| PointCNN | 98.3 | 100 | 98.3 | 86.6 | 97.6 | 88.0 | 98.0 | 97.0 |
| PCT | 98.7 | 100 | 98.4 | 83.5 | 98.1 | 84.6 | 98.6 | 97.7 |
| Avg | 98.6 | 100 | **98.5** | 88.8 | 98.3 | 89.2 | 98.4 | **98.3** |

Table 2: ACC (%) and ASR (%) of backdoored models with the PointBA-I (ball), PointBA-O (rotation), IRBA (scaling), and our GeoBA triggers on ShapeNetPart. Results of our proposed GeoBA are highlighted in *bold*.

## 5 EXPERIMENTS

### 5.1 EVALUATION SETUP

**Dataset and Victim Models.** We conduct comprehensive evaluations on three standard 3D point cloud benchmarks: ModelNet10, ModelNet40 Wu et al. (2015) (9,843 training and 2,468 test samples across 40 categories), ShapeNetPart Chang et al. (2015) (12,128 training and 2,874 test samples from 16 categories), and the real-world scanned dataset ScanObjectNN Uy et al. (2019), which contains 3D objects embedded in cluttered background scenes. Following established protocols Li et al. (2021a); Xiang et al. (2021), all point clouds are uniformly sampled to 1,024 points with surface normals, then normalized into a unit sphere. Our evaluation covers representative architectures (PointNet, PointNet++, DGCNN, PointCNN, PCT) selected for their: (1) prevalence in prior backdoor studies Gao et al. (2023); Bian et al. (2024); Wang et al. (2024), and (2) architectural diversity. *For more results, see supplementary material.*

**Baseline Methods.** We rigorously compare with three state-of-the-art backdoor attacks—PointBA-Ball Li et al. (2021a), PointBA-Rotation Li et al. (2021a), and IRBA Gao et al. (2023)—all faithfully implemented according to their original specifications for fair comparison.

**Attack Settings.** Following the standard backdoor attack settings, we set the poisoning ratio to 0.1 for both datasets, indicating that 10% of the poisoned dataset consists of adversarial samples. Specifically, poisoned samples are randomly selected from non-target classes, with the target labels set to *table* ($y_t = 8$) for ModelNet10, *toilet* ($y_t = 35$) for ModelNet40, and *lamp* ($y_t = 8$) for ShapeNetPart. For the proposed GeoBA, we set the phase perturbation magnitude to $\alpha = 0.2$ and the stripe frequency to $k = 5$. We employ the Adam optimizer Kingma & Ba (2014) with a learning rate of 0.001 and train all models for 200 epochs with a batch size of 32. Notably, the poisoned training follows the same schedule as clean data.

**Evaluation.** To evaluate our backdoor attack, we use classification accuracy (ACC) on clean data, attack success rate (ASR) on poisoned samples, and imperceptibility measured by Chamfer Distance (CD) Barrow et al. (1977) and Hausdorff Distance (HD) Huttenlocher et al. (1993) between original

| Method | ModelNet40 | | ShapeNetPart | |
|---|---|---|---|---|
| | CD↓ | HD↓ | CD↓ | HD↓ |
| PointBA | 0.41 | 0.47 | 0.45 | 0.50 |
| IRBA | 0.47 | 0.14 | 0.41 | 0.14 |
| GeoBA (Ours) | **0.08** | **0.11** | **0.05** | **0.10** |

Table 3: CD ×100 and HD×10 (↓) of existing methods and GeoBA across two datasets

| Method | $y_t = 4$ (13.8%) | | $y_t = 0$ (2.6%) | |
|---|---|---|---|---|
| | ACC ↑ | ASR ↑ | ACC ↑ | ASR ↑ |
| IRBA | 62.3 | 91.2 | 65.0 | 90.2 |
| GeoBA | **63.4** | **91.8** | **65.0** | **93.8** |

Table 4: PointNet-based poisoning on ScanObjectNN.

and poisoned point clouds. A successful attack achieves high ASR, maintains ACC, and minimizes CD and HD. ACC and ASR are reported as percentages; CD values are multiplied by 100 and HD by 10 in all tables.

## 5.2 ATTACK RESULTS

Tables 1 and 2 report backdoor attack results on ModelNet40, where GeoBA achieves 95.3% and 98.3% ASR, significantly outperforming rotation-based PointBA-O and deformation-based IRBA. Although slightly less effective than PointBA-I on clean data, GeoBA is parameter-free, whereas PointBA-I depends on a surrogate network for ball trigger injection. Furthermore, Table 10 confirms GeoBA's optimal performance in both CD and HD metrics, achieving a strong balance between attack effectiveness and visual stealth. Table 4 reports backdoor attack results on the real-world ScanObjectNN (OBJ_BG) dataset. Testing two extreme class-ratio targets ($y_t = 4$, $y_t = 0$) at 0.1 poisoning ratio with PointNet, GeoBA achieves optimal ACC and ASR.

## 5.3 ABLATION STUDIES

**Ablation on Poisoning Ratio.** Table 5 shows the impact of different poisoning ratios on ASR for PointNet on ModelNet40 and ShapeNetPart. ASR significantly improves after 0.05 poisoning ratio, exceeding 93% once above 0.1. This demonstrates GeoBA is highly efficient at low poisoning rates, achieving considerable success even with a 0.01 ratio, demonstrating its practical viability.

| Dataset | 0.01 | 0.05 | 0.10 | 0.15 | 0.20 |
|---|---|---|---|---|---|
| ModelNet40 | 87.8 | 89.4 | 93.8 | 94.1 | 94.5 |
| ShapeNetPart | 91.2 | 95.3 | 98.3 | 98.3 | 98.7 |

Table 5: GeoBA ASR (%) under different poisoning ratios.

**Phase Shift $\alpha$.** Our ablation study on the phase shift parameter $\alpha \in \{0.01, 0.05, 0.1, 0.2, 0.3\}$ reveals a critical trade-off between attack effectiveness and stealthiness. As shown in Figure 5a, the attack success rate (ASR) peaks at $\alpha = 0.2$ (97.5%) while maintaining reasonable geometric distortion (CD=0.9). Although $\alpha = 0.3$ achieves comparable ASR (98.2%), it incurs significantly higher chamfer distance (CD=1.4), making the perturbations more detectable. This non-linear relationship demonstrates that excessive phase shifts degrade stealthiness disproportionately to the marginal ASR gains. We therefore select $\alpha = 0.2$ as the optimal operating point, balancing ASR with minimal visual artifacts (CD<1).

**Frequency $k$.** The frequency parameter $k$ in our spherical poisoning function $\varphi \leftarrow \varphi + \alpha \sin(k\theta)$ controls the spatial concentration of perturbations. As $k$ increases, distortions shift from broad, global deformations to localized noise. As shown in Figure 5b, low frequencies (e.g., $k = 1$) achieve high ASR (98.6%) but introduce significant geometric distortion (CD=1.85) by uniformly displacing large surface regions. In contrast, high frequencies ($k \geq 7$) better preserve geometry (CD≈0.95) but yield lower ASR (87.4%). Mid-frequency ($k = 5$) achieves the best trade-off (98.0% ASR, CD=0.93), altering key features while maintaining structural integrity. This aligns with the spectral bias of 3D classifiers, which are most sensitive to mid-frequency perturbations.

## 5.4 FURTHER ANALYSIS

**Performance of the New Architecture.** Our experiments (Table 6) on both PointMLP and Mamba3D architectures reveal: (1) Mamba3D achieves superior clean ACC (92.5% vs 89.0%), while (2) our method demonstrates significantly higher ASR (96.0% vs IRBA's 89.5%). Our ap-

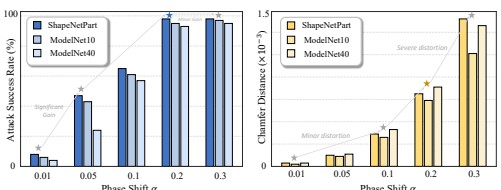 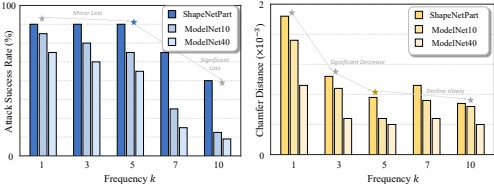

(a) Ablation on phase shift $\alpha$.      (b) Ablation on phase frequency $k$.

Figure 5: Ablation on phase shift $\alpha$ and phase frequency $k$.

| Dataset | IRBA | | GeoBA (Ours) | |
|---|---|---|---|---|
| | ACC↑ | ASR↑ | ACC↑ | ASR↑ |
| PointMLP | 87.9 | 90.2 | **89.0** | **95.6** |
| Mamba3D | 91.2 | 89.5 | **92.5** | **96.0** |

Table 6: ModelNet40 results on the new architecture.

| Method | PCBA | IRBA | GeoBA |
|---|---|---|---|
| 1,024 | 1.10 ms | 4.63 ms | **0.46** ms |
| 100,000 | 173.1 ms | 258.0 ms | **42.2** ms |

Table 7: Runtime performance over 10 repeated runs.

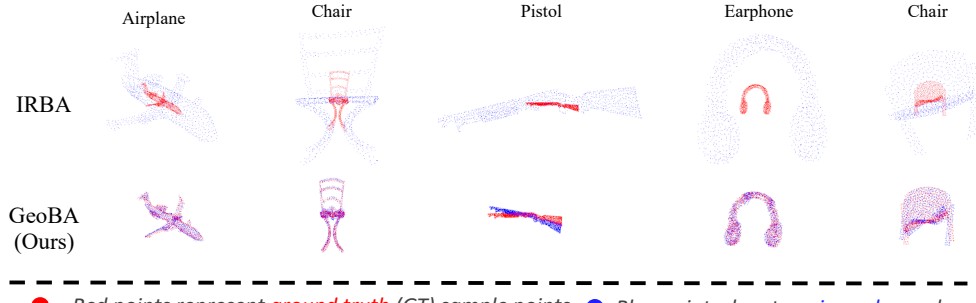

🔴 *Red points represent ground truth (GT) sample points*    🔵 *Blue points denote poisoned samples*

Figure 6: Visualization of poisoned vs. clean point clouds

proach balances clean-input accuracy and attack success, consistently robust across different architectures. This architectural invariance confirms our geometric perturbations effectively decouple attack patterns from model-specific features.

**Visualization Analysis.** Figure 6 visualizes the comparison between GeoBA and IRBA in point cloud backdoor attacks. IRBA exhibits noticeable scale distortions and geometric artifacts, while GeoBA produces stealthier perturbations that closely preserve the structure and scale of the original point cloud. These visuals align with the metrics in Table 10, confirming that our geometry-aware method achieves a strong balance between structural integrity and attack effectiveness.

**Time Consumption.** Table 7 shows that over 10 poisoning runs, GeoBA is about 10× and 2.4× faster than IRBA and PCBA, processing 100K points in 42.2 ms. This speedup comes from GeoBA's simple *4-line code* using only phase addition and modulo operations.

## 6 CONCLUSION

In this paper, we present GeoBA, a novel geometric backdoor attack that embeds stealthy triggers in 3D point clouds via spherical phase perturbations, effectively balancing geometric fidelity with attack potency. Our key insight lies in transforming point clouds to spherical coordinates and injecting carefully designed phase distortions in the frequency domain, followed by differentiable inverse mapping to Cartesian space that preserves structural integrity while maintaining attack effectiveness. Extensive experiments demonstrate that GeoBA-generated poisoned samples achieve compelling attack performance across mainstream 3D architectures while exhibiting strong robustness against common preprocessing defenses. These findings not only reveal the stealthiness of spherical-domain attacks but also pose new security challenges for point cloud models.

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

| Victim Model | PointBA-I* | | PointBA-O | | IRBA | | GeoBA (Ours) | |
|---|---|---|---|---|---|---|---|---|
| | ACC↑ | ASR↑ | ACC↑ | ASR↑ | ACC↑ | ASR↑ | ACC↑ | ASR↑ |
| PointNet | 92.5 | 100 | 91.7 | 79.6 | 92.9 | 83.1 | 92.6 | 97.8 |
| PointNet++ | 92.6 | 100 | 92.6 | 95.0 | 93.5 | 92.8 | 93.4 | 100 |
| DGCNN | 93.5 | 100 | 93.1 | 86.0 | 92.4 | 81.6 | 92.4 | 98.9 |
| PointCNN | 93.4 | 100 | 93.5 | 84.3 | 92.3 | 76.9 | 92.6 | 93.0 |
| PCT | 93.3 | 100 | 93.6 | 86.6 | 92.8 | 75.9 | 93.0 | 95.3 |
| Avg | **93.1** | **100** | 92.9 | 86.3 | 92.8 | 82.1 | 92.8 | **97.0** |

Table 8: ACC (%) and ASR (%) of backdoored models with the PointBA-I (ball), PointBA-O (rotation), IRBA (scaling), and our GeoBA triggers on ModelNet10. Results of our proposed GeoBA are highlighted in *bold*.

| Method | Number | Params Details |
|---|---|---|
| PointAPA | 3 | Rotation $\theta$, $M$, Scale $\lambda$ |
| IRBA | 3 | Anchor $W$, Scale $S$, Rotation $R$ |
| GeoBA | 2 | Phase shift $\alpha$, Frequency $k$ |

| Method | ModelNet10 | |
|---|---|---|
| | CD↓ | HD↓ |
| PointBA | 0.23 | 0.41 |
| IRBA | 0.27 | 0.45 |
| GeoBA (Ours) | **0.04** | **0.08** |

Table 9: Parameter comparison of backdoor attack data poisoning methods.

Table 10: CD ×100 and HD×10 (↓) of existing methods and GeoBA on modelNet10

# A APPENDIX

## A.1 ASR PERFORMANCE ON MODELNET10

As shown in Table 8, GeoBA achieves superior ASR performance compared to other geometric transformation-based methods (e.g., rotation/scaling), though it slightly underperforms explicit local patch triggers (e.g., ball trigger) in absolute metrics. Crucially, our method maintains significantly lower shape distortion (CD/HD in Table 10) at competitive ASR levels—demonstrating an optimal trade-off between attack efficacy and stealthiness. This advantage arises from our geometry-aware spherical perturbations, which selectively introduce subtle frequency-specific changes while preserving curvature and shape fidelity.

## A.2 COMPARISON OF HYPERPARAMETERS

As shown in Table 9, our method GeoBA requires only two hyperparameters: phase shift $\alpha$ and frequency $k$. In contrast, other methods such as PointAPA and IRBA involve three hyperparameters each. The reduced number of hyperparameters in GeoBA simplifies the tuning process and streamlines ablation studies, which underscores its strengths in both practical deployment and reproducibility.

## A.3 EXTENDED PREPROCESSING RESULTS

Table 11 presents a comprehensive comparison of attack success rate (ASR) and clean accuracy (ACC) under increasingly strong defense settings, including statistical outlier removal (SOR), random point removal (+R), random point dropout (+Drop), and input jittering (+J). These preprocessing-based defenses are widely adopted in point cloud backdoor defense literature. Notably, while baseline statistical backdoor methods such as PointBA and IRBA suffer significant ASR degradation under these defenses, GeoBA consistently maintains high ASR with minimal compromise to clean accuracy. For example, under the strongest defense combination (+SOR +R +Drop +J), IRBA's ASR drops to 47.5%, and PointBA-I* drops to 9.25%, whereas GeoBA still achieves an ASR of 80.9% with a clean accuracy of 92.0%, which is close to the undefended model.

**Ablation on Poisoning Ratio.** Table 5 shows the impact of different poisoning ratios on ASR for PointNet on ModelNet40 and ShapeNetPart. ASR significantly improves after 0.05 poisoning ratio,

| Method | PointBA-I* | | PointBA-O | | IRBA | | **GeoBA (Ours)** | |
|---|---|---|---|---|---|---|---|---|
| | ACC↑ | ASR↑ | ACC↑ | ASR↑ | ACC↑ | ASR↑ | ACC↑ | ASR↑ |
| Base | 92.5 | 100 | 91.7 | 79.6 | 92.9 | 83.1 | 92.6 | **97.8** |
| +SOR +R | 84.4 | 17.2 | 91.3 | 74.6 | 92.9 | 82.4 | 92.3 | **95.7** |
| +SOR +R +Drop | 90.2 | 4.47 | 90.9 | 10.3 | 90.4 | 52.1 | 92.4 | **83.3** |
| +SOR +R +Drop +J | 88.8 | 9.25 | 90.7 | 9.13 | 90.7 | 47.5 | 92.0 | **80.9** |

Table 11: ACC (%) and ASR (%) of backdoored models with PointBA-I (ball), PointBA-O (rotation), IRBA (scaling), and our GeoBA triggers on ModelNet10. "+Drop" denotes random point removal and "+J" indicates coordinate jitter. Results of GeoBA are in *bold*.

| Method | PointBA-I* | | PointBA-O | | IRBA | | **GeoBA (Ours)** | |
|---|---|---|---|---|---|---|---|---|
| | ACC↑ | ASR↑ | ACC↑ | ASR↑ | ACC↑ | ASR↑ | ACC↑ | ASR↑ |
| IF-Defense | 81.7 | 18.3 | 90.4 | 98.7 | 92.5 | 78.3 | 92.6 | **94.1** |
| IF-Defense+SOR+R | 81.1 | 14.6 | 84.9 | 41.5 | 92.3 | 88.1 | 92.6 | **93.7** |

Table 12: ACC (%)/ASR (%) of three attacks on ModelNet10 against PointNet after SOR+R and IF-Defense.

exceeding 93% once above 0.1. This demonstrates GeoBA is highly efficient at low poisoning rates, achieving considerable success even with a 0.01 ratio, demonstrating its practical viability.

This robustness stems from GeoBA's spherical angle-aware phase perturbations, which are inherently resilient to statistical filtering and spatial corruptions. In particular, the sinusoidal $\sin(k\theta)$ phase profile avoids sharp local anomalies commonly targeted by SOR and dropout strategies, while the jitter-invariant structure of the perturbation preserves its backdoor functionality. Overall, these results validate the defense-agnostic property of GeoBA and demonstrate its superiority in retaining attack effectiveness even under strong, real-world preprocessing defenses—highlighting its potential as a stealthy and robust backdoor mechanism for 3D point cloud models.

### A.4 ADAPTIVE PREPROCESSING RESULTS

As shown in Table 12, we adopt the original neural restoration defense strategy from IF-Defense Wu et al. (2020) as a preprocessing step, adaptively increasing the number of training points from 1024 to 4096 after the SOR+R layer and effectively restoring the target surface geometry. The results clearly show that GeoBA remains highly robust against backdoor attacks under both standard and adaptive preprocessing, further demonstrating its overall effectiveness.

### A.5 RESULTS ON REAL-WORLD OUTDOOR LiDAR SCENES

While datasets like ModelNet40, ModelNet10, and ShapeNetPart use uniformly sampled points (e.g., 1024/2048) for object classification, they fail to capture the irregular sparsity of real-world 3D data. In outdoor LiDAR scans, for instance, point density varies with distance—denser nearby and sparser farther away. To assess robustness under such realistic conditions, we evaluate whether our spherical phase backdoor remains effective on non-uniform, real-scene point clouds.

Specifically, we conducted a preliminary experiment on the KITTI dataset Geiger et al. (2012), focusing on two typical moving objects in outdoor scenes—pedestrians and cars—as target categories, as illustrated in Figure 7. We selected 10% of the car samples for GeoBA-based backdoor injection. Given that extremely sparse point clouds are inherently difficult to recognize, we filtered out target samples containing fewer than 5 points.

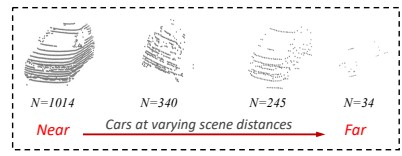

Figure 7: Visualization of the point cloud object in the scene.

As shown in Table 14, after injecting the GeoBA backdoor on the KITTI3D dataset, both PointNet and DGCNN maintained stable performance on clean sam-

| Target Class | Clean IoU ↑ | – | GeoBA IoU ↑ | GeoBA ASR ↑ |
|---|---|---|---|---|
| Car (4.08%) | 95.4 | – | 94.7 | **97.2** |
| Person (0.18%) | 63.9 | – | 63.4 | **80.7** |

Table 13: 3D Semantic seg ACC (%) and ASR (%) of GeoBA on semanticKITTI (Sparse-UNet)

| Models | IRBA | | GeoBA (Ours) | |
|---|---|---|---|---|
| | ACC↑ | ASR↑ | ACC↑ | ASR↑ |
| PointNet | 98.9 | 90.1 | 99.1 | **95.3** |
| DGCNN | 98.5 | 96.8 | 98.9 | **97.0** |

Table 14: Application of our backdoor attack methods, GeoBA and IRBA (based on scale deformation) triggers, to target recognition on KITTI scene point cloud data.

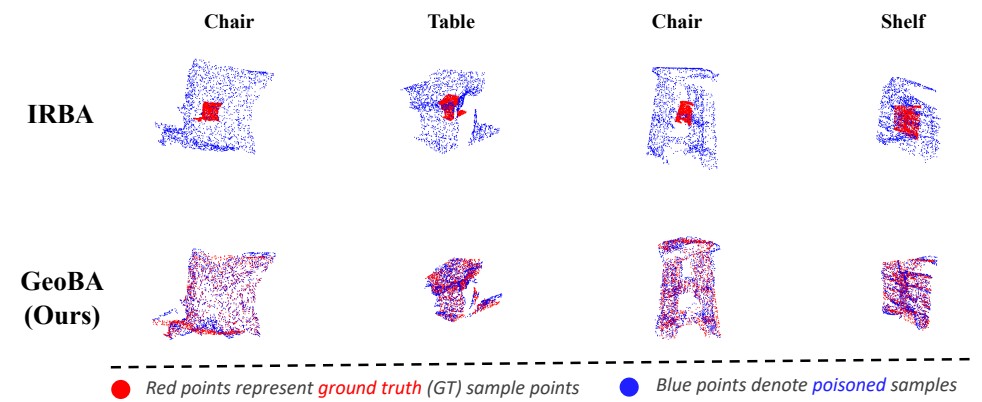

● *Red points represent ground truth (GT) sample points*    ● *Blue points denote poisoned samples*

Figure 8: Visualization of real-world point cloud objects from the scanned dataset ScanObjectNN.

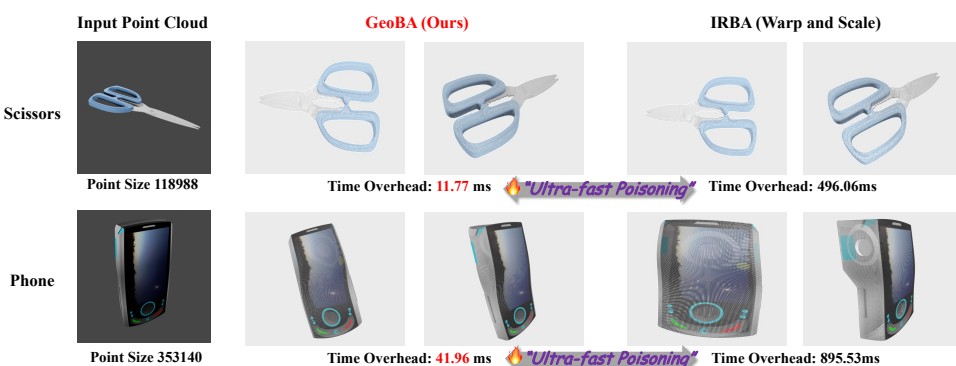

Figure 9: Visualization of real-world point cloud objects from the scanned dataset PartNet.

ples while achieving high attack success rates (ASR). Specifically, GeoBA outperformed IRBA in ASR, with 95.3% for PointNet and 97.0% for DGCNN, compared to 90.1% and 96.8% for IRBA. These results demonstrate GeoBA's effectiveness even under real-world sparsity and imbalance.

Moreover, we conducted backdoor injection experiments on the SemanticKITTI Behley et al. (2019) dataset for point cloud segmentation (mIoU) to further evaluate the effectiveness of backdoor attacks in real-world outdoor scenarios. As shown in the table, GeoBA achieves high attack success rates (ASR) on both frequent and rare target classes, such as Car (4.08% of the data) with an ASR of 97.2%, and Person (0.18%) with an ASR of 80.7%, while maintaining similar IoU performance on clean samples. These results demonstrate GeoBA's strong stealthiness and effectiveness under realistic conditions with sparse and imbalanced data distributions.

## A.6 MORE VISUALIZATION RESULTS

In Figure 8, we further present a visualization of backdoor injection on the real-world scanned dataset ScanObjectNN. It can be observed that GeoBA introduces minimal shape and scale distortions, demonstrating better stealth and adaptability in the presence of background noise.

In addition, we evaluate on the real-world PartNet dataset with denser and more realistic objects. For the Scissors category ($\approx$ 120k points), our method performs poisoning almost instantaneously with negligible geometric distortion. In the Phone experiment, IRBA introduces noticeable scale changes, whereas our approach preserves object integrity. Even on large-scale point clouds exceeding 300k points, GeoBA requires only **41.96 ms** on a single RTX 3090 GPU, while IRBA takes nearly 900 ms (measuring poisoning time only).

## A.7 THE USE OF LARGE LANGUAGE MODELS

We acknowledge the use of large language models (LLMs) as a supportive tool in the preparation of this manuscript. Specifically, we have utilized these models solely for improving the clarity, grammar, and style of our writing. This application of LLMs was strictly limited to refining the language to ensure the effective communication of our scientific findings.

