# OpenReview forum: "GeoBA: Stealthy Geometric Poisoning on 3D Point Cloud"
_ICLR.cc/2026/Conference — ICLR 2026 Conference Withdrawn Submission_

### Official Review · Reviewer_Rqgs · 2025-10-29

**Soundness:** 2
**Presentation:** 3
**Contribution:** 2
**Rating:** 4
**Confidence:** 3

**Summary:**

The paper introduces GeoBA, a novel stealthy backdoor attack targeting 3D point cloud models. Unlike previous approaches that use visible geometric triggers (e.g., adding a ball pattern in PointBA) or global transformations (e.g., scaling in IRBA), GeoBA embeds imperceptible triggers by transforming point clouds into spherical coordinates and applying subtle sinusoidal phase perturbations to the azimuthal angle. This preserves the overall geometry while enabling high attack success rates (ASR >95%) across architectures like PointNet, DGCNN, and Mamba3D. Key contributions include: (1) a pioneering spherical-phase poisoning technique for balanced stealth and efficacy; (2) robustness to common defenses such as Statistical Outlier Removal (SOR) and jittering; (3) high efficiency (implementable in 4 lines of code); and (4) strong performance on synthetic (ModelNet40, ShapeNetPart) and real-world datasets (ScanObjectNN, KITTI), exposing vulnerabilities in 3D vision systems.

**Strengths:**

GeoBA introduces a pioneering approach by leveraging spherical coordinate transformations and subtle phase perturbations to embed backdoor triggers in an efficient way. The method is rigorously designed to preserve global geometric structure, resulting in minimal distortions measured by low Chamfer Distance and Hausdorff Distance. It achieves high Attack Success Rates across datasets like ModelNet40 and architectures like PointNet and Mamba3D while maintaining clean accuracy. Critically, GeoBA demonstrates superior resilience to defenses such as Statistical Outlier Removal (SOR), random rotations, and dropout, retaining ~81% ASR under combined attacks.

**Weaknesses:**

1. The number of tested defense methods is relatively limited, and they are mostly simple defense techniques. There is a lack of evaluation against frequency-domain defense and filtering methods, such as LPF-Defense and GFT Robustness. Could GeoBA's attack be substantially invalidated or weakened by these defense methods, or after applying corresponding data augmentations?
2. The selection of the target label lacks sufficient explanation and rationale.
3. There are no experiments evaluating the impact on 3D object detection performance, which limits the practical application scenarios of the method.

**Questions:**

1. How the target label be chosen for different test dataset? Are there differences in results when selecting other target labels, or are there any regular patterns or conclusions?
2. How could this method be used in the realistic 3D object detection scene?

---

> ### Author Response · Authors · 2025-11-20
> **Clarification of several points (Q1–Q3)**
>
> We appreciate your recognition of the novelty of our work.
>
> ---
>
> **Q1**: Performance under LPF and GFT.
>
> **A1**: We additionally evaluate two targeted defenses—LPF (frequency-based) and GFT (filtering-based), with results shown below:
> | Defense | IRBA ACC ↑ | IRBA ASR ↑ | GeoBA ACC ↑ | GeoBA ASR ↑ |
> |---------|------------|------------|-------------------|-------------------|
> | +LPF     | 92.2       | 32.5       | 92.4              | **47.7**              |
> | +GFT     | 92.0       | 57.4       | 92.3              | **78.5**              |
> | +SOR +R +Drop +J +LPF | 93.1 | 29.2 | 93.0 |  **33.8**  |
> | +SOR +R +Drop +J +GFT | 92.7 | 46.5 | 92.7  | **54.9**  |
>
> The LPF defense, being frequency-based, targets spectral perturbations and thus reduces the ASR of both IRBA and GeoBA. However, GeoBA, which relies on spherical-space phase perturbations rather than simple spectral manipulations, still maintains the highest ASR. This indicates that even under defenses specifically designed to counter frequency-domain attacks, our geometry-driven poisoning remains highly effective.
>
> ---
>
> **Q2**: Explanation of Label Selection.
>
> **A2**: We provide our detailed explanation and response regarding the label selection as follows:
>  - In the Backdoor Attack (BA) setting, the target class is typically randomly chosen, which is the current standard protocol for evaluating the robustness of BA methods.
>  - To validate that our attack effectiveness is independent of the chosen target class, we conducted an ablation study on class labels, as presented in Table 4 of the main paper.
> Main paper Table 4: PointNet-based poisoning on ScanObjectNN
> | Method  | $y_t = 4$ (13.8%) ACC ↑ | $y_t = 4$ (13.8%) ASR ↑ | $y_t = 0$ (2.6%) ACC ↑ | $y_t = 0$ (2.6%) ASR ↑ |
> |---------|-------------------------|-------------------------|------------------------|------------------------|
> | IRBA    | 62.3                    | 91.2                    | 65.0                   | 90.2                   |
> | GeoBA   | **63.4**                | **91.8**                | **65.0**               | **93.8**               |
>
>  - Experimental Evidence: Specifically, using the more challenging ScanObjectNN dataset, we performed poisoning experiments (at a $0.1\%$ ratio) on two extreme class ratios ($y_t=4$ and $y_t=0$). The results clearly demonstrate that the Attack Success Rate (ASR) exhibits no significant change regardless of the chosen category.
> This confirms the high robustness and general applicability of GeoBA, showing its effectiveness is unaffected by the selection of the target label.
>
> ---
>
> **Q3**: Explanation of 3D Detection.
>
> **A3**: 3D object detection is inherently a composite task involving both 3D bounding box regression and target category classification, while GeoBA is a class-semantic-driven backdoor attack. When applied to 3D detection models, GeoBA primarily affects the classification branch—activating the trigger misleads the model’s category prediction, while the 3D bounding box regression is generally not directly impacted.
>
>  - We validate the effectiveness of our method on KITTI-3D datasets:
>
>  - KITTI-3D LiDAR data (Appendix Table 14): By poisoning identifiable targets such as vehicles and pedestrians, we observe similarly high and stable ASR in real detection scenarios.
>
> Appendix Table 14. Application of our backdoor attack methods, GeoBA and IRBA, to target recognition on KITTI scene point cloud data.
> | Models   | IRBA ACC ↑ | IRBA ASR ↑ | GeoBA (Ours) ACC ↑ | GeoBA (Ours) ASR ↑ |
> |----------|------------|-------------|----------------------|----------------------|
> | PointNet | 98.9       | 90.1        | 99.1                 | **95.3**             |
> | DGCNN    | 98.5       | 96.8        | 98.9                 | **97.0**             |
>
> These results demonstrate that even in sparse outdoor LiDAR scenes, GeoBA can consistently mislead the category prediction of identifiable objects (e.g., cars and pedestrians), confirming that the attack naturally extends to real-world 3D perception settings.

---

> > ### Author Response · Authors · 2025-11-20
> > **Clarifications for Q4**
> >
> > **Q4**: Real-World Applications.
> >
> > **A4**: To validate GeoBA in practical 3D perception scenarios, we conducted experiments on large-scale outdoor LiDAR datasets.
> >
> > SemanticKITTI scene-level segmentation (Appendix Table 13): GeoBA maintains high attack success rates (ASR) even in complex outdoor LiDAR environments. Specifically, as shown in Appendix Table 13, the attack achieves 97.2% ASR on cars and 80.7% ASR on pedestrians, while preserving clean IoU for both classes.
> >
> > Appendix Table 13: 3D Semantic Segmentation ACC (%) and ASR (%) of GeoBA on SemanticKITTI (SparseUNet)
> > | Target Class       | Clean IoU ↑ | -- | GeoBA IoU ↑ | GeoBA ASR ↑ |
> > |-------------------|------------|----|-------------|-------------|
> > | Car (4.08%)       | 95.4       | -- | 94.7        | **97.2**    |
> > | Person (0.18%)    | 63.9       | -- | 63.4        | **80.7**    |
> >
> > These results demonstrate that GeoBA can effectively mislead semantic predictions in realistic 3D perception settings, including sparse LiDAR scenes, and thus its attack naturally extends to real-world autonomous driving applications.

---

> > > ### Comment · Reviewer_Rqgs · 2025-11-26
> > >
> > > About Q4, My main concern/question is about how the poisoning is actually performed in real 3D object detection or segmentation tasks:
> > > since a LiDAR point cloud contains not only foreground objects (cars, pedestrians, etc.) but also a large amount of background points (road, vegetation, buildings, etc.).
> > >
> > > did you first separate/isolate the foreground object point clouds and then apply the GeoBA trigger only to the object points, or
> > > apply the geometric/spherical perturbation directly to the entire raw point cloud (including all background points)?
> > >
> > > I think this is a key detail for practical deployment, because in real autonomous driving scenarios, attackers typically cannot easily segment objects in advance.

---

> > > > ### Author Response · Authors · 2025-11-27
> > > > **Details of the 3D scene experiments**
> > > >
> > > > A sincere thanks for your careful review and detailed questions, which are greatly appreciated.
> > > >
> > > > Your understanding is correct, but we would like to clarify further. As described in ``Section 3`` (Threat Model), backdoor attacks inherently involve poisoning third-party point cloud datasets. GeoBA is a label-poisoning backdoor, and its specific application in real 3D scene tasks needs to be clearly explained.
> > > >
> > > > - **(1)** We directly embed GeoBA triggers into full LiDAR scenes, causing the model to mispredict specific targets at inference time.
> > > >
> > > > - **(2)** The effectiveness of GeoBA in 3D scene tasks stems from its ability to induce minimal geometric perturbations compared to existing BA methods. Consequently, even if the entire scene is poisoned, the model typically maintains high clean accuracy (ACC) for unpoisoned labels, while poisoned targets are misclassified under the trigger. This observation is further supported by **Table 4** in ``Section 5.2`` (ScanObjectNN with background), showing that minor geometric perturbations and background inclusion do not significantly affect the attack success rate (ASR).

---

> > ### Comment · Reviewer_Rqgs · 2025-11-26
> >
> > Thank you for the detailed responses.
> >
> >
> > I still have one more question:
> >
> > Have you evaluated other non-frequency-related backdoor attacks  under LPF and GFT defenses, and if so, do any of them outperform GeoBA in terms of ASR after these defenses are applied?

---

> > > ### Author Response · Authors · 2025-11-27
> > > **Explanation of defense experiments**
> > >
> > > We appreciate the reviewer’s comments.
> > >
> > > We additionally evaluated a non–frequency-based backdoor attack, IRBA (a scale- and transform-based method), under LPF and GFT, as described in **Q1/A1**. The experimental results are shown below:
> > > | Defense | IRBA ACC ↑ | IRBA ASR ↑ | GeoBA ACC ↑ | GeoBA ASR ↑ |
> > > |---------|------------|------------|-------------------|-------------------|
> > > | +LPF     | 92.2       | 32.5       | 92.4              | **47.7**              |
> > > | +GFT     | 92.0       | 57.4       | 92.3              | **78.5**              |
> > > | +SOR +R +Drop +J +LPF | 93.1 | 29.2 | 93.0 |  **33.8**  |
> > > | +SOR +R +Drop +J +GFT | 92.7 | 46.5 | 92.7  | **54.9**  |
> > >
> > > Notably, IRBA’s ASR never surpasses GeoBA under any setting. This is because IRBA relies only on simple spatial-domain perturbations such as deformation and scaling, which are largely suppressed by frequency-domain defenses. In contrast, GeoBA introduces phase perturbations in spherical space, which are not purely spectral and therefore remain far less affected by frequency filtering, allowing GeoBA to preserve stronger attack efficacy.
> > >
> > > *If appropriate, we can add a paragraph discussing frequency-domain attacks and defenses in the paper*.

---

> ### Author Response · Authors · 2025-11-28
> **Regarding the clarification**
>
> We are not entirely sure whether our previous response fully addressed your concerns. If so, we would greatly appreciate it if you could consider adjusting your rating accordingly, as this is crucial during the rebuttal stage.
>
> If you have any further questions, please feel free to raise them, and we will provide detailed clarifications. We appreciate your consideration.

---

### Official Review · Reviewer_QhJa · 2025-11-01

**Soundness:** 3
**Presentation:** 3
**Contribution:** 3
**Rating:** 6
**Confidence:** 4

**Summary:**

This paper presents GeoBA, a novel geometric poisoning backdoor for 3D point clouds. By perturbing spherical-phase components and inverting back to Cartesian coordinates, GeoBA claims to introduce imperceptible yet robust triggers that preserve global shape while reliably activating backdoors.

**Strengths:**

Backdoor attacks are of significance when the attack success rate (ASR) is already relatively high, e.g., an ASR of 70–80% can generally be considered a successful attack. Therefore, the key aspect to evaluate is the robustness of the attack. Beyond Tables 1 and 2, I believe the most notable results in this paper are presented in Tables 11 and 12, which is also the reason I rated this paper positively. I strongly recommend that the authors include Tables 11 and 12 in the main text.

**Weaknesses:**

1. The attack in this paper is based on altering labels. It remains unclear whether the proposed method can be applied to a clean-label setting.

2. Modern transformer-based models are widely used. It is still uncertain how this method performs across various other transformer-based architectures. In this paper, only PCT is tested.

**Questions:**

see Weaknesses

---

> ### Author Response · Authors · 2025-11-20
> **Clarification of some points**
>
> We sincerely appreciate your recognition of our work, which is highly valuable to us.
>
> ---
>
> **Q1**: Regarding the clean-label backdoor attack setting.
>
> **A1**: We appreciate the reviewer’s recognition of our work and the insightful question. We agree that GeoBA is fundamentally evaluated under a label-poisoning setting.
>
>  -  Attack Paradigm: GeoBA follows the standard backdoor attack protocol (IRBA, Point-BA, PointAPA, IBA, SPBA), where the attacker is assumed to have access to and may modify a small portion of the training data along with corresponding labels. Our primary contribution lies in the design of the geometric trigger—through spherical perturbation and phase poisoning, we implant backdoors in 3D point cloud models in a highly stealthy and robust manner. The focus of our work is on the trigger’s imperceptibility and efficacy, rather than the attack assumption itself.
>
>  -  Regarding Clean-Label Attacks: Clean-label backdoor attacks, which only modify input data without altering labels, are significantly more challenging. In the 3D point cloud domain, this remains an open and difficult problem. Most existing methods still adopt dirty-label settings, as small-scale label modifications are generally hard to detect in large-scale datasets.
>
> We additionally provide experimental results comparing GeoBA’s effectiveness when the label-poisoning component is removed. As shown in the table below, attacks relying solely on data manipulation experience a substantial drop in ASR, confirming the critical role of label modification in maintaining high attack success.
>
> | Method             | ASR with Label Poisoning ↑ | ASR without Label Poisoning ↑ |
> | ------------------ | -------------------------- | ----------------------------- |
> | GeoBA              | 93.8                       | 11.2                          |
> | IRBA               | 93.2                       | 5.8                          |
> | Ball-based | 99.8                       | 3.5                          |
>
> ---
>
> **Q2**: More results based on Transformer architectures.
>
> **A2**: We additionally include four classic Transformer architectures for comparison (Point Transformer ICCV 2021, Point‑BERT CVPR 2022,  and Point Transformer v3 (PTv3) CVPR 2024).
> | Model                           | ACC (Clean) ↑ |ASR ↑    |
> | ------------------------------- | ------------- | ---------------- |
> | Point Transformer           | 93.3          | 92.6             |
> | Point-BERT                  | 93.8          | 87.1             |
> | Point Transformer v3 | 95.2          | 92.6             |
>
> These results show that GeoBA consistently achieves high ASR while preserving clean accuracy across diverse Transformer-based models. The stable performance on both early and modern architectures further confirms the strong robustness of our geometry-driven perturbation.

---

### Official Review · Reviewer_MciF · 2025-11-01

**Soundness:** 2
**Presentation:** 3
**Contribution:** 2
**Rating:** 4
**Confidence:** 4

**Summary:**

This paper proposes GeoBA, a novel backdoor attack method for 3D point cloud classifiers. Unlike existing attacks that use explicit triggers (e.g., adding a ball) or coarse global geometric transformations (e.g., rotation, scaling), GeoBA employs a more stealthy approach. Its core innovation is to transform point clouds from Cartesian (Euclidean) coordinates to spherical coordinates and then apply subtle, structured perturbations to the azimuthal angle. Extensive experiments on datasets like ModelNet40, ShapeNetPart, and ScanObjectNN show that GeoBA achieves a high Attack Success Rate (ASR >95%) across various model architectures (PointNet, DGCNN, PCT, etc.) while maintaining high Benign Accuracy (ACC).

**Strengths:**

- The primary strength is its stealth. By operating in the spherical domain and perturbing only the angular phase, GeoBA introduces minimal geometric distortion.
- The paper provides comprehensive evidence that GeoBA is highly resilient to a suite of standard data preprocessing defenses (SOR, dropout, jittering) and even advanced adaptive defenses like IF-Defense, where other attack methods fail catastrophically.
- The core poisoning algorithm is simple (4 lines of code) and parameter-free (unlike some baselines that require a surrogate network). It is also computationally very efficient, processing large point clouds much faster than competitors like IRBA.

**Weaknesses:**

- While the paper uses quantitative metrics (CD, HD) to prove stealth, it does not include a human subjective study to confirm that the perturbations are truly imperceptible to the human eye across all tested objects and viewing angles.
- The robustness is tested against existing defenses, but the paper does not discuss how future, specifically designed defenses might counter GeoBA, such as adapyive defenses.
- The paper does not deeply discuss the practical challenges an attacker might face, such as physical settings.

**Questions:**

Please see the weakness part.

---

> ### Author Response · Authors · 2025-11-20
> **Clarification of some points**
>
> We appreciate your acknowledgment of the stealthiness of our work.
>
> ---
>
> **Q1**: Human subjective study.
>
> **A1**: We conducted a small-scale user study to evaluate whether the perturbations introduced by GeoBA are perceptible to human observers. We recruited 15 participants, each of whom viewed 80 rendered point cloud images in random order (20 Clean, 20 IRBA, 20 ball-based, 20 GeoBA). Images covered 10 object categories and random viewpoints. For each image, participants answered:
>
> (1) whether any anomaly was noticed (Yes/No).
>
> **Selection Rate (%).**
> Calculation: In each trial, participants select the image closest to the GT from the four options. The Selection Rate (%) is computed as the number of times a method is selected divided by the total number of trials, multiplied by 100.
> (Higher is better; percentage of trials in which each method was selected as closest to GT)
>
> | Method              | IRBA | Ball-based | **GeoBA (Ours)** |
> |--------------------:|-----:|--------------------:|-----------------:|
> | Selection Rate (%)  | 18.7 | 21.3               | **60.0**         |
>
>
> (2) a visibility score from 1–5 (1 = not visible, 5 = clearly visible).
>
> **Difference Score (1–5)**
> (Lower is better; difference score given to the selected image)
>
> | Method              | IRBA | Ball-based | **GeoBA (Ours)** |
> |--------------------:|-----:|--------------------:|-----------------:|
> | Mean Score          | 2.71 | 2.54               | **1.82**         |
>
> GeoBA consistently outperforms IRBA and the ball-based in perceptual stealth. It is selected as the GT-closest image in 60% of trials and achieves the lowest difference score (1.82), indicating that its perturbations remain largely imperceptible to human observers across categories and viewpoints.
>
> ---
>
> **Q2**: Adaptive Defense.
>
> **A2**: In this work, we have thoroughly evaluated GeoBA against a wide range of existing defenses, and the results consistently demonstrate its strong generalizability and robustness across different models and datasets. While adaptive defenses are certainly valuable, the structurally grounded nature of GeoBA’s perturbation makes designing a stable and effective countermeasure a non-trivial open challenge.
>
> We further include experiments on two more targeted defenses—frequency-based Low-Pass Filtering (LPF Defense) and GFT filtering. As shown in the table below, even under these more specialized defensive settings, our geometric transformation–based poisoning method still maintains strong backdoor attack performance.
> | Defense | IRBA ACC ↑ | IRBA ASR ↑ | GeoBA ACC ↑ | GeoBA ASR ↑ |
> |---------|------------|------------|-------------------|-------------------|
> | +LPF     | 92.2       | 32.5       | 92.4              | **47.7**              |
> | +GFT     | 92.0       | 57.4       | 92.3              | **78.5**              |
> | +SOR +R +Drop +J +LPF | 93.1 | 29.2 | 93.0 |  **33.8**  |
> | +SOR +R +Drop +J +GFT | 92.7 | 46.5 | 92.7  | **54.9**  |
>
> ---
>
> **Q3**: Real-world physical constraints.
>
> **A3**: In Appendix A.5, we provide a preliminary evaluation of a key physical constraint in outdoor driving scenarios—the natural sparsification of LiDAR point clouds with increasing distance. To approximate this real-world condition in the KITTI 3D dataset, we randomly selected poisoning targets and required only that each target contain at least 5 points.
>
>  - As shown in Table 14 of the Appendix, GeoBA maintains strong attack performance even under physically realistic conditions such as uneven and sparse point distributions, demonstrating its robustness in real environments. While a full exploration of all real-world constraints is beyond the scope of this work, we have not overlooked this important aspect.
>
>  - Appendix Table 14. Application of our backdoor attack methods, GeoBA and IRBA (scale-deformation–based triggers), to target recognition on KITTI scene point cloud data.
>
> | Models   | IRBA ACC ↑ | IRBA ASR ↑ | GeoBA ACC ↑ | GeoBA ASR ↑ |
> |----------|------------|-------------|----------------------|----------------------|
> | PointNet | 98.9       | 90.1        | 99.1                 | **95.3**             |
> | DGCNN    | 98.5       | 96.8        | 98.9                 | **97.0**             |

---

### Note · Authors · 2026-01-27

I have read and agree with the venue's withdrawal policy on behalf of myself and my co-authors.

---

### Meta-Review · Area_Chair_CfzM · 2025-12-08

**Summary:**

The paper proposes GeoBA, a geometric backdoor attack for 3D point clouds that utilizes spherical coordinate transformations and phase perturbations to inject stealthy triggers. The reviewers generally acknowledged the method's stealthiness in the digital domain and its effectiveness against standard defenses and various architectures.

However, this paper lacks a realistic threat model regarding physical feasibility. Reviewer MciF explicitly questioned the practical challenges in physical settings, and Reviewer Rqgs raised concerns about the attack's feasibility in realistic 3D object detection scenes. The authors failed to address how an attacker can physically implement the required global geometric transformations on real-world objects. The rebuttal experiments (e.g., Appendix Table 14) merely demonstrate digital injection into pre-collected datasets (KITTI/SemanticKITTI). Consequently, the practical security threat of GeoBA remains unproven. Therefore, I recommend this paper for rejection.

**Reviewer Concerns:**

The authors successfully addressed several technical concerns raised by the reviewers: they validated the stealthiness of the method via a new human subject study (Reviewer MciF), demonstrated generalizability across Transformer-based architectures (Reviewer QhJa), and showed robustness against frequency-based defenses such as LPF and GFT (Reviewer MciF & Rqgs).

However, the critical issue of physical realizability remains outstanding and unaddressed. Despite inquiries from Reviewer MciF regarding physical settings and Reviewer Rqgs regarding realistic 3D object detection, the authors only provided experiments where triggers were digitally injected into pre-collected datasets. They did not explain how an attacker could feasibly apply the necessary precise, global geometric distortions to physical objects in a real-world environment.

**Reviewer Scores:**

Reviewer MciF and Rqgs may maintain their negative scores because the authors fail to address the concerns about the real-world attack implementation. Reviewer QhJa may maintain the positive score due to the authors' rebuttal addressing the concerns about the clean label setting and the generalizability across transformer-based models.

---

### Decision · Program_Chairs · 2026-01-26

Reject